



# Contribution of wood burning to exposures of PAHs and oxy-PAHs in Eastern Sweden

Hwanmi Lim[1], Sanna Silvergren[4], Silvia Spinicci[2], Farshid Mashayekhy Rad[3], Ulrika Nilsson[1], Roger Westerholm[1], Christer Johansson[4,5]

[1]Department of Materials and Environmental Chemistry, Stockholm University, Stockholm, 106 91, Sweden
[2]Waters Sverige AB, 171 65, Solna, Sweden
[3]Department of Chemistry, Uppsala University, Box 576, 751 23 Uppsala, Sweden
[4]Environment and Health Administration, SLB, Stockholm, 104 20, Sweden
[5]Department of Environmental Science, Stockholm University, Stockholm, 106 91, Sweden

*Correspondence to*: Christer Johansson (Christer.Johansson@aces.su.se)

## Abstract

A growing trend in the developed countries is the use of wood as fuel for domestic heating due to measures taken to reduce usage of fossil fuels. However, this imposed another issue with the environment and human health. That is, the emission from wood burning contributed to the increased level of atmospheric particulates and the wood smoke caused various respiratory diseases. The aim of this study was to investigate the impact of wood burning on the polycyclic aromatic hydrocarbons (PAH) in air $PM_{10}$ using known wood burning tracers, i.e. levoglucosan, mannosan and galactosan from the measurement at the urban background and residential areas in Sweden. A yearly measurement from three residential areas in Sweden showed a clear seasonal variation of PAHs during the cold season, mainly from the increased domestic heating and meteorology. Together, an increased sugar level assured the wood burning during the same period. The sugar ratio (levoglucosan/(mannosan+galactosan)), was a good marker for wood burning source such as wood type used for domestic heating and garden waste burning. On the Walpurgis Night, the urban background measurement demonstrated a dramatic increase in levoglucosan, benzo[*a*]pyrene (B[*a*]P), and oxygenated PAHs (OPAH) concentrations from the increased wood burning. A significant correlation between levoglucosan and OPAHs was observed, suggesting OPAHs to be an indicator of wood burning together with levoglucosan. The levoglucosan tracer method and modelling used in predicting the B[*a*]P concentration could not fully explain the measured levels in the cold season. The model showed that the local wood source contributed to 98% of B[*a*]P emissions in Stockholm area and 2% from the local traffic. However, non-local sources were dominating in urban background (60%). A further risk assessment estimated that the airborne particulate PAHs caused 13.4 cancer cases per 0.1 million inhabitants in Stockholm County.


## 1 Introduction

In the year 2016, fuel combustion, especially biomass for domestic heating, was the single most important source of airborne benzo[*a*]pyrene (B[*a*]P) as well as many other polycyclic aromatic hydrocarbons (PAH), primary fine particulate matter (PM) and black carbon (BC) in Europe (EEA, 2018a). Concentrations of airborne PM of particle diameters less than 10 or 2.5 μm

(PM$_{10}$ and PM$_{2.5}$) on a daily or yearly basis and B[*a*]P on annual basis is regulated by the EU Ambient Air Quality Directives. For daily mean PM10 concentrations a maximum of 35 days per year is allowed to exceed 50 μg m$^{-3}$ (WHO, 2006; EU, 2008; EEA, 2017). Most incidents with exceedances of the daily PM10 limit value due to the increased residential heating from wood burning occur during the cold season (Maenhaut et al., 2012). During the winter, wood smoke has been estimated to contribute to approximately 20 – 50 % of the total organic matter (Puxbaum et al., 2007) and contribute to elevated levels of PM10,

PM2.5, B[a]P and BC and (Yttri et al., 2005; Favez et al., 2009; Maenhaut et al., 2012; Wagener et al., 2012; Fuller et al., 2014; Beekmann et al., 2015; Cordell et al., 2016; Glasius et al., 2018). The use of wood as a fuel for domestic heating is growing in the industrialised countries as an alternative to fossil fuels. Between 2010 and 2020 a 57–110 % increase of biomass usage has been estimated as a result of CO$_2$ emission related policies and regulations (Wagner et al., 2010).

Air pollution from biomass burning has been estimated to contribute to at least 40,000 premature deaths in Europe (Sigsgaard

et al., 2015). Earlier studies have shown that exposure to wood smoke is associated with respiratory diseases, e.g. asthma (Boman et al., 2003), and possibly also cardiovascular diseases (Kocbach Bølling et al., 2009). It was also noted that different combustion conditions may lead to different types of wood smoke particles with varying physicochemical properties and health risks (Kocbach Bølling et al., 2009).

B[*a*]P is regarded as a marker for PAHs, which originate from different types of incomplete combustion of organic matter,

such as fossil and biomass fuels and smoking (Boström et al., 2002; EU, 2005, 2008; Lewtas, 2007). B[*a*]P has been classified as a known human carcinogen (Group I) by the International Agency for Research on Cancer (IARC), which recently added outdoor air pollution and PM to the same group (IARC, 2010 and 2016). There are also other PAHs classified as probable and possible human carcinogens (IARC 2010), but among PAHs it is only B[*a*]P that is regulated (annual mean of 1 ng m$^{-3}$ as target value) in the EU Ambient Air Quality Directives. Additionally, a compound belonging to the class of oxygenated PAHs

(OPAH), 9,10-anthraquinone, has been classified as possible human carcinogen (Group 2B) (IARC, 2013), but is still not on the regulation list. The main emission source of OPAHs are the traffic and biomass burning, but they are also formed at photo degradation of PAHs (Alves et al., 2017). The OPAHs are mainly particle-associated in the atmosphere due to their low vapour pressure (Walgraeve et al., 2010; Delgado-Saborit et al., 2013).

A highly selective tracer for burning of wood is levoglucosan (1,6-anhydro-β-D-glucopyranose), a monosaccharide derivative

formed when cellulose is pyrolysed at high temperatures (Shafizadeh, 1968; Simoneit et al., 1999). Due to the high concentration in the smoke and a high chemical stability, this tracer compound can be detected in the atmosphere through long-range transports (Simoneit et al., 1999; Fraser and Lakshmanan, 2000). In addition, mannosan and galactosan released from the thermal degradation of hemicellulose are also detected in wood smoke emissions and suggested to be source-specific tracers





for wood burning (Nolte et al., 2001; Simoneit, 2002). Either levoglucosan, or the combination of all three monosaccharides, was measured to estimate the contribution of wood burning to the total air PM (Yttri et al., 2005; Caseiro et al., 2009; Maenhaut et al., 2012, 2016; Wagener et al., 2012; Glasius et al., 2018).

A significant reduction in the emissions of air pollutants has been observed in Europe since 1990 due to measures taken in the energy, road transport and industry area, as well as fuel change and advances in fuel technology, e.g. desulphurisation (EEA 2018b). $PM_{10}$ and $PM_{2.5}$ have undergone a considerable decrease in emissions from the public electricity and heat production while a steady state or even an increasing trend was observed in the residential category (EEA 2018b). B[$a$]P, on the other hand, showed first a clear declining trend until 2000, but since then a gradual increase has been shown (EEA 2018b). The emission inventory report, however, noted that PAH emission was mainly from the chemical industry (42 %) whereas the commercial, institutional and households including residential sources are the main emission category for $PM_{10}$ and $PM_{2.5}$, contributing with 39 % and 56 %, respectively (EEA, 2018b), which implies that the emissions of both $PM_{10}$ and $PM_{2.5}$ from households are increasing.

The aim of the present study was to quantify the importance of wood burning for PAHs, OPAHs and $PM_{10}$ during different seasons in three different residential areas in Sweden and the impacts of biomass burning on the urban background during the episodic events using tracers of wood burning. Altogether, the lung cancer risk of air PM associated with the exposure was estimated in the urban background and residential areas.

## 2 Methods

### 2.1 Chemicals and solvents

All solvents used were of HPLC grade: toluene, hexane and methyl *tert*-butyl ether from Rathburn Chemicals (Walkerburn, UK), and acetonitrile from VWR (Radnor, PA, USA). Anhydrous dodecane (> 99 %) was purchased from Sigma-Aldrich (St Louis, MO, USA). PAHs, OPAHs and internal standards (ISs) are listed in Table S1. Supplier and purity information of PAH and OPAH standards, and ISs are given elsewhere (Ahmed et al., 2015; Nyström et al., 2016). Levoglucosan (1,6-anhydro-β-D-glucose) of 99 % purity was obtained from Sigma-Aldrich (St. Louis, MO, USA) and its isotope-labelled IS, levoglucosan-$^{13}C_6$ of 98 % purity was from Cambridge Isotope Laboratories, Inc. (Andover, MA, USA). Mannosan (1,6-anhydro-β-D-mannose) and galactosan (1,6-anhydro-β-D-galactose), both of more than 95 % purity, were purchased from Cayman chemical (Ann Arbor, MI, USA). A Milli-Q plus system from Millipore (Bedford, MO, USA) was used to prepare ultra-pure water with a resistivity of 18 MΩ·cm.





## 2.2 Sampling

### 2.2.1 Sampling location characterisation

Three different residential sampling sites in Sweden were selected to collect $PM_{10}$ filter samples, as well as one rural site for measurements of the non-local contributions, as shown in Fig. 1. In detail, Delsbo (DE, N 61.80378, E 16.55225) is a low-

5      traffic villa area in the western Hudiksvall, a city approx. 300 km north of Stockholm. The other two residential areas are located in Stockholm, Enskede (EN, N 59.28004, E 18.04046) and Södertälje, Ytterjärna (YJ, N 59.08761, E 17.57143) with road traffic influence. The rural background sample was collected in Aspvreten (ASP, N 58.80584, E 17.38832), approximately 80 km southwest of Stockholm. In addition, two sampling stations close to Enskede in central Stockholm (not shown in the map) were also included, one located in a densely trafficked street canyon, Hornsgatan (HG, N 59.31728, E 18.04984) and

10     one on a roof-top (24 m above the streets) classified as an urban background site, Torkel Knutssonsgatan (TK, N 59.31605, E 18.05785). Basic meteorological parameters (temperature, wind) and other air quality data (black carbon, $NO_x$, $PM_{10}$, and $PM_{2.5}$) for these two areas were available from the Environment and Health Administration (SLB), Stockholm (http://slb.nu/slbanalys/).

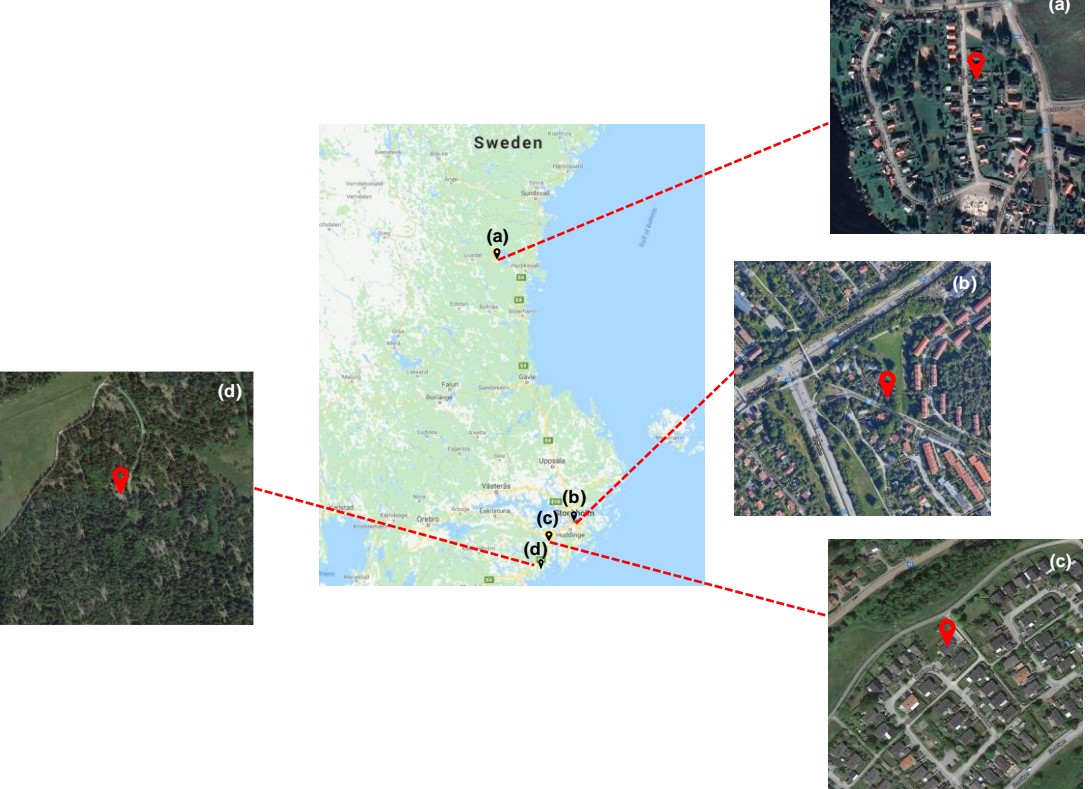

15     **Figure 1**. Geographical location of four sampling sites in Sweden: **(a)** Delsbo (DE), **(b)** Enskede (EN), **(c)** Ytterjärna (YJ), and **(d)** Aspvreten (ASP). Map data: ©Google, DigitalGlobe, and Lantmäteriet/Metria.



### 2.2.2 PM$_{10}$ sampling

Particulate matter (<10 µm aerodynamic diameter) was collected by active sampling through a micro-quartz fibre filter (Ø = 47 mm, T293 grade, Munktell, Ahlstrom-Munksjö, Stockholm, Sweden) using a sequential sampler (SEQ47/50, Sven Leckel, Ingenieurbüro GmbH, Berlin, Germany) at an airflow rate of 1 m$^3$ h$^{-1}$. The sampling period was set to 24 h from Jan to Apr, after which it was set to 48 h until the end of the year 2017. A sampling campaign was done at TK, the urban background site, from Feb 22 to May 6 in 2016, thereby including the Walpurgis Night event (Apr 30). The samples were collected in the same way as the other sampling sites, but with a flow rate of 2.3 m$^3$ h$^{-1}$. A field blank was prepared using a filter treated the same as a sample except that no air was taken via sampler. The individual filter sample was stored at -20 ℃ until extraction.

### 2.3 Sample preparation

The filter sample was put at room temperature for 1 h after taken out from the fridge, then cut into three parts. The first and second quarters were used for PAH, OPAH (TK sample only) and sugar analysis while the remaining half was kept as backup for re-analysis. All three parts were weighed separately. As shown in Table S2, the cut filters were pooled into half calendar months during winter season (Nov-Apr) when higher emission was expected, while whole-month samples were analysed during warmer period (May-Oct).

### 2.3.1 PAH

The filter samples were prepared as described previously (Ahmed et al., 2015). In brief, the pooled filters were first spiked with ISs and then extracted using an accelerated solvent extraction (ASE) system. The extracts were then applied to a silica SPE column and eluted with toluene. The final eluate was subjected to LC-GC/MS analysis. All analytical parameters are given in detail elsewhere (Sadiktsis et al., 2014; Olsson et al., 2014). The field and method blank filters were extracted and prepared in the same manner as sample filters. However, the PAH data for rural background samples were provided by the Swedish Environmental Research Institute (https://www.ivl.se).

### 2.3.2 OPAHs

OPAHs were collected on filters at TK during the spring in 2016. The samples were prepared similarly as for the PAH determination, except that an additional IS, deuterated 9,10-anthraquinone (AQ-$d_8$), was spiked onto each filter before the ASE. The details of sample preparation and the parameters used in LC-GC/MS analysis have been described previously (Ahmed et al., 2015). Both field and method blank filters were prepared and analysed in the same way as the samples.



### 2.3.3 Levoglucosna, mannosan and galactosan

In brief, the pooled filters were spiked with $^{13}$C-labelled levoglucosan as IS prior to extraction by ultrasonication in ultrapure water. The extracts were vacuum-dried, then re-dissolved in water/acetonitrile (5:95) and finally analysed by HILIC/ESI-MS/MS. The details of sample preparation and instrumental analysis have been described previously (Mashayekhy Rad et al., 2018). Both field and method blank filters were extracted and prepared in the same manner as the sample filters.

### 2.4 Emission inventory of B[*a*]P

A previous emission inventory during 2010 quantified local sources of B[a]P within Stockholm county and neighbouring Uppsala county for emissions during year 2006 (SLB-analys, 2010). In the inventory local a total of 73 kg of B[a]P was accounted of which 96 % was local traffic and residential wood sources. The remaining sources was residential oil combustion, boilers, industrial energy sources and shipping, which together contributed to 4 % of the emitted B[a]P. Non-local sources, e.g. transport of B[a]P from outside the regions, were concluded to be a substantial source as well. Based upon the previous emission inventory focus was directed towards local traffic and residential wood sources in the presented work.

### 2.4.1 Residential wood combustion

An emission inventory of residential wood combustion was built including 0.28 million objects based on chimney sweeper documentations and housing data in Stockholm County (see Appendix for details). The methodology was to a large extent based on a report from the Swedish Meteorological and Hydrological Institute (SMHI) (Andersson et al., 2018).

### 2.4.2 Road traffic

We used a detailed bottom up inventory of traffic in Stockholm County, part of the Eastern Swedish Air Quality Association (in Swedish: Östra Sveriges Luftvårdsförbund). The traffic exhaust emissions in the inventory are based on data from The Handbook Emission Factors for Road Transport, (HBEFA) 3.3 (http://www.hbefa.net/e/index.html). It includes emission factors for passenger cars (PC), light duty vehicles (LDV, < 3.5 ton), heavy goods vehicles (HGV, > 3.5 ton), urban buses, coaches and motorcycles, each divided into different categories, for a wide range of traffic situations. As in a previous publication on B[*a*]P emissions from wood in Helsinki (Douros and Moussiopoulos, 2014; Hellén et al., 2017) a factor of 0.000031 was applied on PM exhaust emissions to estimate the B[*a*]P contribution from traffic.

### 2.4.3 Non-local sources

The emission inventory only covers local sources of B[*a*]P from residential wood burning and road traffic. However, there is transport from outside regions that affects the concentrations of airborne B[*a*]P in Stockholm County. There are several regional background sites located in areas with few inhabitants where continuous monitoring of B[*a*]P and other air pollutants



are measured. In present work we have used regional background data for 2014-2018 from Råö, a coastal area in Kungsbacka in Western Sweden. It is located upwind from the other sampling locations in this work. There is another background site available located in a forested area just outside Stockholm, but it was dismissed due to interference and local emissions showing higher concentrations of B[*a*]P than those in the suburban and urban measurements. The B[*a*]P concentration in Råö was very

similar in 2017 (0.042 ng m$^{-3}$) as the average during 2014-2018, with concentrations ranging between 0.033-0.053 ng m$^{-3}$ during the period.

### 2.4.4 Model simulations

All dispersion modelling of local emissions was performed with a Gaussian plume model within in the Airviro air quality management system (Apertum, 2021; Segersson et al., 2017). Airviro uses a diagnostic wind model based on the theory first

described by Danard (Danard, 1977), in which it is assumed that small scale winds can be seen as a local adaptation of large-scale winds. Calculation of B[*a*]P concentrations was performed for local wood and traffic sources in a 5.5 km$^2$ size area with a 100 m grid size around the measurement sites using meteorology during the measurement period from a 50 m meteorological mast located in Högdalen, a suburb in Stockholm. Modelled monthly mean concentrations due to local residential wood combustion and road traffic plus the non-local source contributions (based on regional background measurements) was

compared with monthly measured concentrations. Dispersion modelling was also done for entire Stockholm County for local wood and local traffic sources using climatology for 1993-2010. The climatology includes horizontal and vertical wind speed, wind direction, temperature, temperature difference between three levels and also global radiation. The data is grouped into 60 wind direction classes with 6 stability classes within each wind sector. The wind model also takes varying local topographical properties into account. The measured concentration of B[*a*]P in regional background was added to the

calculated local amount and the total B[*a*]P concentration can then be compared with the measured concentration during the measurement campaign in Stockholm urban background (TK), residential area (EN) and (YJ). The measurement in the northern residential area (DE) is outside of the emission inventory area and cannot be compared with calculated concentrations.

### 2.5 B[*a*]P equivalent concentration

B[*a*]P equivalent concentrations (B[*a*]P$_{eq}$) were calculated by multiplying individual PAH concentration with its relative

potency factor (RPF). As shown in Table 1, this study used 17 PAHs for the total B[*a*]P$_{eq}$.

Table 1. The relative potency factor (RPF) for PAHs used in this study.

| PAH | RPF | PAH | RPF | PAH | RPF | PAH | RPF | PAH | RPF | PAH | RPF |
|---|---|---|---|---|---|---|---|---|---|---|---|
| B[*a*]P[a] | 1 | B[*a*]A[a] | 0.2 | B[*k*]F[a] | 0.03 | I[1,2,3-*cd*]P[a] | 0.07 | Anthan[a] | 0.4 | DB[*a,i*]P[a] | 0.6 |
| Flu[a] | 0.08 | 5MChr[a] | 1 | B[*j*]F[a] | 0.3 | DB[*a,h*]A[a] | 10 | DB[*a,l*]P[a] | 30 | DB[*a,h*]P[a] | 0.9 |
| B[*c*]f[a] | 20 | B[*b*]F[a] | 0.8 | B[*j*]A[b] | 30 | B[*ghi*]p[a] | 0.009 | DB[*a,e*]P[a] | 0.4 | | |

[a] MDH, 2016, [b] Lim et al., 2015. See Table S1 for compound abbreviations.

## 3 Results

### 3.1 Seasonal variation of PAH and sugar levels

The monthly variations of the PAHs and sugars in 2017 are shown in Fig 2 a and b, respectively. In January the most northern area (DE) showed much higher levels of both PAHs and sugars and also lower temperatures than the two areas in Stockholm (EN and YJ). This is not only due to increased emissions, but also meteorology. That is, the low temperature is associated with higher emissions from residential heating, traffic, and less efficient ventilation of pollutants due to lower wind speeds and more stable atmosphere, involving more frequent situations with inversions. There were three occasions when the sugar levels were increased more than the PAHs, indicating the increased emission source of levoglucosan, mannosan and galactosan, i.e. wood burning. For all sampling sites, the lowest levels are detected in the summertime.

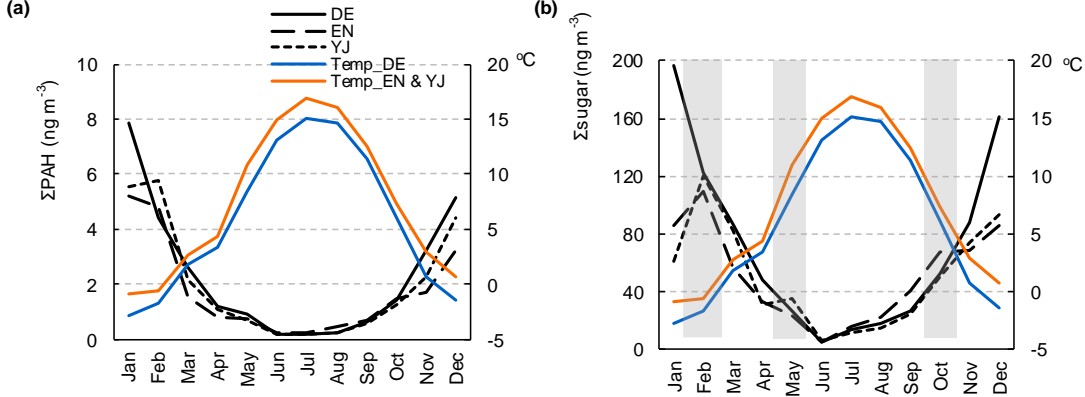

**Figure 2**. Seasonal variation of **(a)** ΣPAHs (sum of 48 PAHs in Table S1) and **(b)** Σsugars (sum of levoglucosan, mannosan and galactosan) in $PM_{10}$ collected from three sampling sites (DE, EN and YJ) in 2017. The local temperature data was obtained from the Swedish Meteorological and Hydrological Institute (https://www.smhi.se). Periodic occasions with elevated sugar level were marked in grey colour in **(b)**.

### PAHs

The PAH concentrations showed a strong seasonal variation at all three sites as shown in Fig. 3. On the left side, the mass concentration of low and high molecular weight PAHs (LMW and HMW) are compared together with those of the total PAHs. PAHs with three and four rings were grouped in LMW while five- and six-ring PAHs were in HMW. The pie chart (right side in Fig. 3) shows the relative abundance of LMW and HMW during summer (Jun-Jul) and winter (Jan-Feb). A considerable shift in the relative LMW level from winter to summer was observed in all three locations with the largest change in DE. The distinctive seasonal shift observed in DE was also from the increased residential heating, which mostly affected the PAHs with





four rings. The increased emission of four-ring PAHs from domestic heating was also reported in the high Arctic during winter (Singh et al., 2017). The sugar levels, in general, followed the seasonal variation as the PAHs, however, there were occasions with increased sugar levels when biomass burning or wood combustion happened.

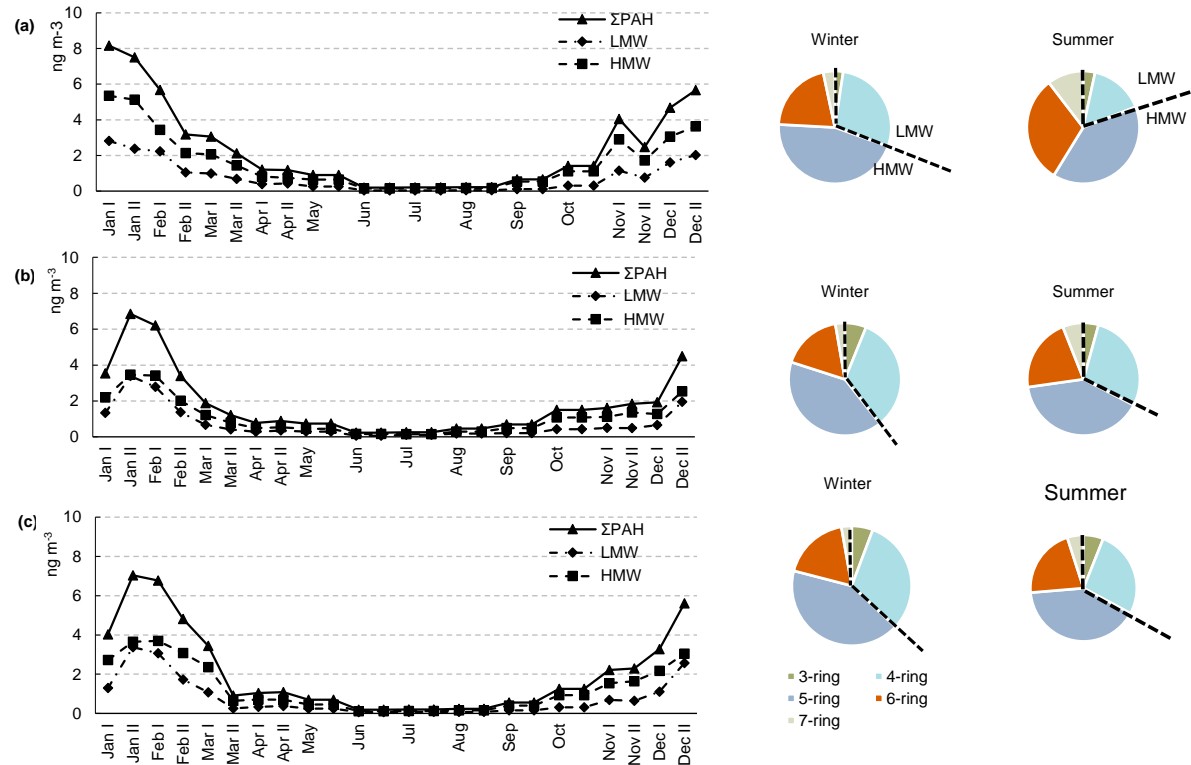

**Figure 3**. Seasonal variation of low and high molecular weight PAHs in PM$_{10}$ collected from three sampling sites in 2017: **(a)** DE, **(b)** EN and **(c)** YJ.

**Levoglucosan, mannosan and galactosan**

Annual sugar concentrations from three sampling locations are summarised in Fig. 4. The level is higher during colder season, January in DE, and February in EN and YJ while the level was lowest in June from all sampling sites. The seasonal variation of levoglucosan concentrations was the largest in DE (4.96-167 ng m$^{-3}$), followed by EN and YJ (4.41-105 and 5.30-111 ng m$^{-3}$, respectively) as shown in Table S3.





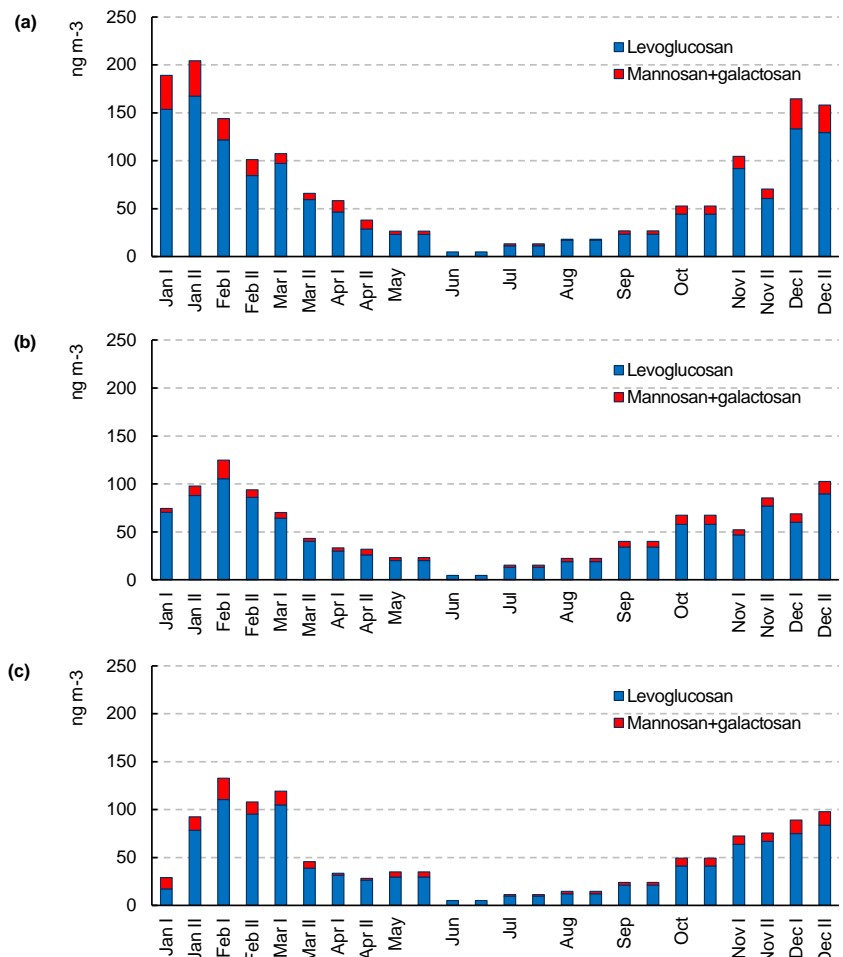

**Figure 4**. Distribution of annual measurement data for levoglucosan, mannosan, and galactosan from three sampling locations. Mannosan and galactosan are determined as sum due to co-elution in the chromatogram.

**3.2 PAH and sugar emission trend associated with residential wood burning**

Other air quality parameters, e.g. $NO_x$, $PM_{10}$ and $PM_{2.5}$, were also measured at the urban background site (TK) during the sampling period (Fig. S1 in S1). The seasonal variation was the greatest at the residential area in the northern Sweden (DE) with regard to the winter-to-summer ratio of B[$a$]P concentration (Table S4).

The variation in both total PAH concentration and the sum of anhydrous monosaccharides (levoglucosan, mannosan and galactosan) is shown in Fig. 5. Also, the relative ratio between levoglucosan and the other two isomers (LG/(M+G)) is plotted to estimate the contribution of different wood sources to the sugar emission. The hard wood burning generally gave higher ratios (8.5-17.6) while combustion from soft wood showed lower values (1.8-3.6) (Nolte et al., 2001; Fine et al., 2002, 2004;



Schmidl et al., 2008). The reported sugar ratios in the air PM ranged from 3.2 to 14 (Fabbri et al., 2009) and this study showed comparable but slightly wider sugar ratios (1.46-18.6).

Detailed results and discussion for each location are as follows:

**Delsbo**

A clear winter-to-summer variation in the sugar emission was observed in the northern villa area as shown in Fig. 5 (a). The measured sugar level in Jan was almost twice as high as those from the other two locations while the PAH levels were similar in all three locations throughout the year. It is likely that there was a substantial increase in wood burning in DE during the winter, largely from the residential heating. The sugar ratio (LG/(M+G)) during the winter varied within the range 4.3-5.5, suggesting that both hard and soft wood was used as fuel for heating. During spring there were two occasions with elevated sugar levels. The first was in Mar with the indication of hard wood burning (ratio 9.7 and 9.4), while the second was in Apr with mixed wood combustion sources (4.0 and 3.1).

15 **Enskede**

Fig. 5 (b) shows the annual variation in PAH and sugar emissions in the trafficked urban residential area. The winter-to-summer variation was observed, but the sugar emissions were lower than in DE during the cold season, implying less of wood burning. The sugar ratios revealed a predominant burning of hard wood. Apart from the annual event of the Walpurgis Night, Stockholm city allowed residents to burn their garden wastes (e.g. grasses, leaves, weeds, shrubs, and wood) twice a year, in weeks 18 (May 1-7, 2017) and 40 (Oct 2-8, 2017). This clearly reflected by the elevated sugar level in October and the sugar ratio (5.2), the latter indicating mixed combustion sources.

**Ytterjärna**

As shown in Fig. 5 (c), the suburban residential area in Stockholm also showed a seasonal variation in PAH and total sugar similar to DE and EN. However, a different trend was observed with the sugar ratios. It was noted that the sugar emissions were largely from mixed wood burning, except for Apr when the sugar ratios were as high as 16.8 and 13.3, indicating hard wood burning. This infers that the residents in YJ used both hard and soft wood for domestic usage, but predominantly hard wood for special occasions, e.g. Walpurgis Night. The elevated sugar level in May was related to the garden waste burning week, as reflected by the sugar ratio of 5.6.



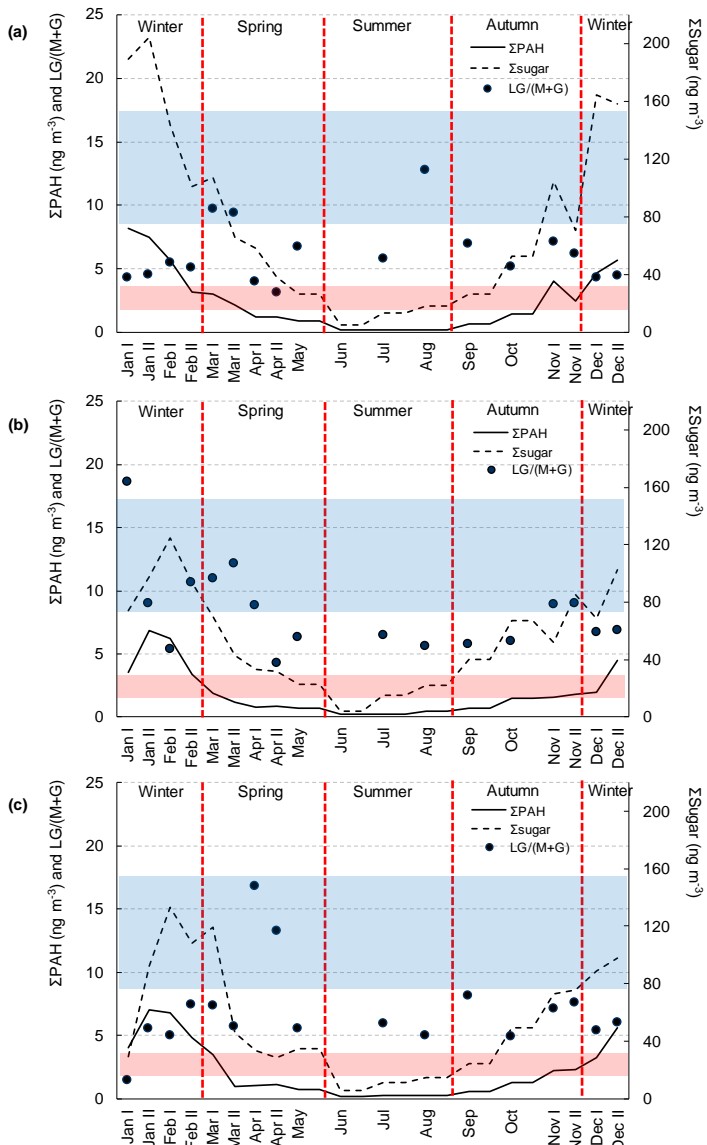

**Figure 5.** Seasonal variation of ΣPAHs, Σsugars, and levoglucosan/(mannosan+galactosan) (LG/(M+G)) ratio in PM$_{10}$ collected from three sampling sites in 2017: **(a)** DE, **(b)** EN and **(c)** YJ. The blue and red zone represents the previously reported sugar ratio from hard and soft wood burning, respectively.

### 3.3 Impact of non-residential biomass burning

The measured levels of PAHs, OPAHs and sugars in airborne PM$_{10}$ taken at the urban background (TK) around the Walpurgis Night (Apr 30) in 2016 are shown in Fig. 6. A clear elevation of both the total sugar and levoglucosan can be observed in the sample collected on the event day. In the present study, the total sugar level showed an increase of almost three times, from




36.4 to 117 ng m$^{-3}$. The total OPAH level was also shown to increase approximately threefold. The sugar ratios (4.4-7.2) from the urban background indicated a mixed emission source of hard and soft wood combustion, except for the sample collected on the 30$^{th}$ Apr. This sample exhibited sugar ratio of 8.8, i.e. reflecting hard wood burning. Detailed information is given in Table S5.

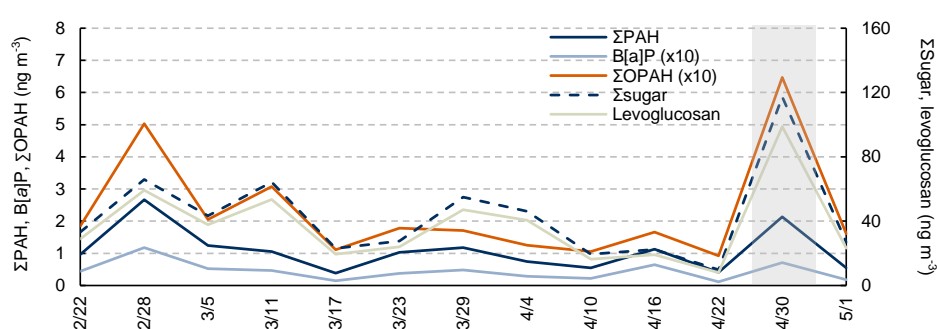

**Figure 6.** Measured concentrations of ΣPAHs, B[$a$]P, ΣOPAH, Σsugars and levoglucosan in PM$_{10}$ collected from Stockholm urban background (TK) during Feb 22 – May 5, 2016. In **(a)** the red line represents the upper limit sugar ratio from soft wood burning while the blue line was the lower value from hard wood burning. The grey area indicates the measurements of the
Walpurgis Night sample.

**3.4 Modelled concentrations of B[$a$]P**

The modelled results at all measurement sites compared to measured data are shown in Fig 7 and 8. At all three locations, urban background (TK) and residential areas (YJ, EN), the average modelled B[$a$]P concentration is higher than the measured
value. The modelled concentrations are overestimated consistently, both during year and at all sites. The poor correlation between modelled and measured data is likely to have several causes. Regional background data is very important to add to the modelled concentrations, but it also adds to the uncertainty, partly depending on the prevailing wind direction and whether local emission affects the measurements. Also, previous work has showed that emissions vary significantly between old and new wood stoves (Omstedt et al 2007), details that was unavailable in the chimney sweeper data.  In this work we used
emissions factors based on Todorovic et al. (2007), EMEP/EEA (2013), SMED and the Swedish EPA Naturvårdsverket (2017). The range of emission factors documented in Todorovic (2007) is shown in Table A4 in Appendix, clearly showing the difficulty in predicting B[$a$]P concentrations by modelling. For example, an emission factor range between 0.004-0.27 (0.05 was used in inventory) was documented for wood stoves, the type of stove that dominate the local wood sources in the Stockholm emission inventory. The assumed emission factor for wood stoves could alone vary the B[$a$]P concentration
outcome significantly.





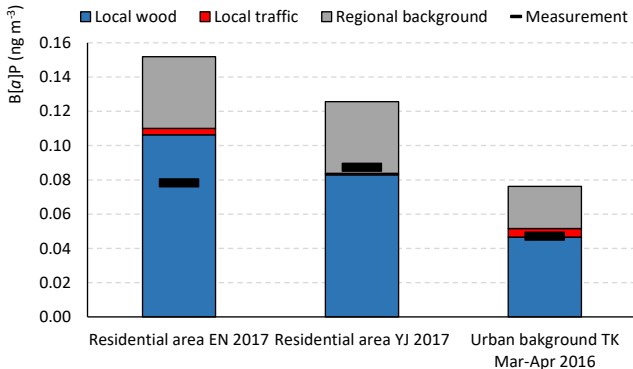

**Figure 7**. Modelled concentrations of B[*a*]P in Stockholm compared with measured values. In urban background (TK), the regional background data are from 29 Feb – 2 May and measured data from 28 Feb – 1 May.





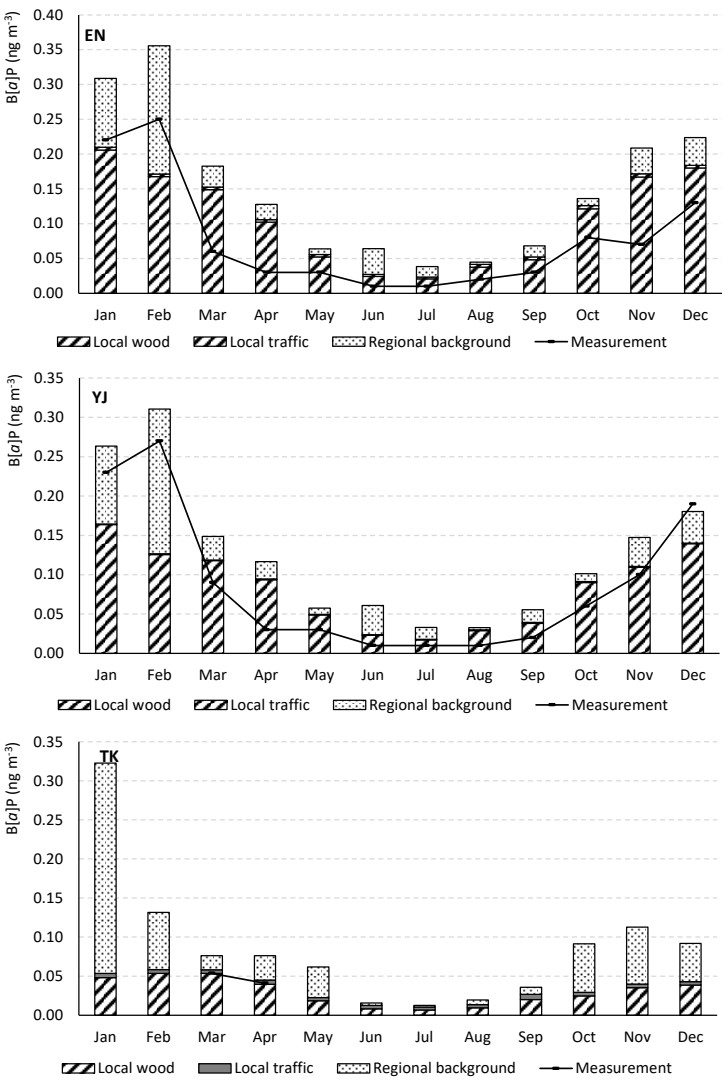

**Figure 8**. Modelled monthly concentrations of B[*a*]P in residential areas (EN and YJ) and urban background (TK) compared with measured values. Measured data is from 2017 for residential areas and from April-May 2016 for urban background.

## 4 Discussion

5 **4.1 Levoglucosan: strong relationship with black carbon, PM$_{2.5}$ and oxy-PAHs**

At the urban background site (TK), the highest correlation ($r_s$) of levoglucosan values was found with biomass BC (0.87), PM$_{2.5}$ (0.86), $\Sigma$OPAH (0.85), and total BC (0.80) followed by other parameters, $\Sigma$PAH (0.75), B[*a*]P (0.70) and PM$_{10}$ (0.63) as shown in Fig. 9. The strong relationship between levoglucosan and PM$_{2.5}$ was also reported for another Northern European city, Oslo (residential, urban background and street area), but not for other European areas such as the Netherlands, Germany





and Spain (Jedynska et al., 2015). The strong correlation with black carbon suggests biomass burning under high temperatures as shown previously. (Saleh et al., 2014; Martinsson et al., 2015). The levoglucosan emission levels are much strongly correlated with ΣOPAH than with ΣPAH and B[*a*]P, which implies the origin of OPAHs in the urban background is mainly biomass burning and wood combustion, while ΣPAH and B[*a*]P originate from other emission sources such as fossil fuel

5   combustion.

**Figure 9.** Scatter plot and Spearman correlation coefficient ($r_S$) of measured meteorological and air quality parameters against levoglucosan level at the urban background (TK). [*]Statistically significant with $p < 0.05$, [**]Statistically significant with $p < 0.01$, [***]Statistically significant with $p < 0.001$.

**4.4 Estimation of source contribution to B[*a*]P**

**Dispersion model**





According to the emission inventory for Stockholm County (approx. 2 million habitants) the annual emission of B[a]P from local residential wood burning is 186 kg. That can be translated into approximately 0.08 g B[a]P per inhabitant. In Helsinki, with 1.1 million inhabitants, 196 kg B[a]P emission was estimated from wood sources in 2014 (Hellén et al., 2017). That is approximately 0.18 g B[a]P per inhabitant. In Finland sauna stoves account for a large amount of the total emission source

and sauna stoves are generally high emitters. Wood sources contribute to approximately 98 % of the total local emission of B[a]P in the inventory for Stockholm County. The remaining 2 % is expected to be from local traffic sources. The calculated annual B[a]P from traffic exhaust in Stockholm County was 4.5 kg based on the year 2015. A very similar result was estimated for Helsinki traffic in 2014. B[a]P from traffic in Stockholm County is approximately 2 % of the total estimated local B[a]P emissions (traffic and wood).

The estimated source contributions to B[a]P from local wood, local traffic and regional background are shown in Table 2. According to the modelled results local residential wood sources account for 60-70 % of the total B[a]P in residential areas (EN) and (YJ). In urban background (TK) traffic sources are relatively more important; 7/8 of B[a]P with local origin is estimated to be from residential heating and 1/8 from traffic source. At all sites regional background is a significant source. In urban background it is expected to account for around 60 %. On average local emissions contribute more than 1/4 of the

average B[a]P concentration in entire Stockholm County, showing wood sources as dominating contributor.

The estimated B[a]P from biomass burning, using levoglucosan as marker, accounts for 33–156 % and 11-221 % of the total B[a]P at the urban background (TK) and residential areas (DE, EN and YJ), respectively (Fig. S2). As seen in Table 2, the model indicates that the sources of biomass burning in the residential areas (EN and YJ) and urban background (TK) are different. The former is mainly from the local wood burning while the latter comes from the combination of local and regional

background.

Table 2. Calculated percentage contribution from local and non-local sources to B[a]P.

| Measurement station | Period | Local wood (%) | Local traffic (%) | Regional background (%) | Biomass burning [a] |
|---|---|---|---|---|---|
| EN (Residential area) | 2017 | 70 | 2 | 27 | 60 |
| YJ (Residential area) | 2017 | 66 | 1 | 33 | 51 |
| TK (Urban background) | 2016 | 34 | 5 | 61 | - [b] |
| TK (Urban background) | Mar – Apr 2016 | 61 | 7 | 32 | 91 |
| Average Stockholm County | 2017 | 26 | 1 | 73 | - [b] |





[a] Biomass burning contribution to B[*a*]P estimated from levoglucosan (B[*a*]P$_{BB}$ = 0.0011 × levoglucosan, $r^2$ = 0.54, Belis et al., 2011), [b] Data not available since the levoglucosan measurement was not done throughout the year in 2016.

**4.5 Modelled concentration fields of B[*a*]P from local sources and total B[*a*]P concentrations in Stockholm County**

All results shown below in Fig. 10 are based on climatology and represents a normal meteorological year. However, the total concentration is based on the 5-year average from measurements during 2014-2018 in Råö. Local wood burning contributes to approximately 98 % of the total local sources from residential wood burning and traffic inventory data. It can be seen in Fig. 10 (a) and (b) that the traffic sources mainly affect the B[a]P concentrations in the central parts of Stockholm, whilst the emissions from local wood sources result in higher concentrations of B[a]P in the rather central suburban regions of Stockholm County.

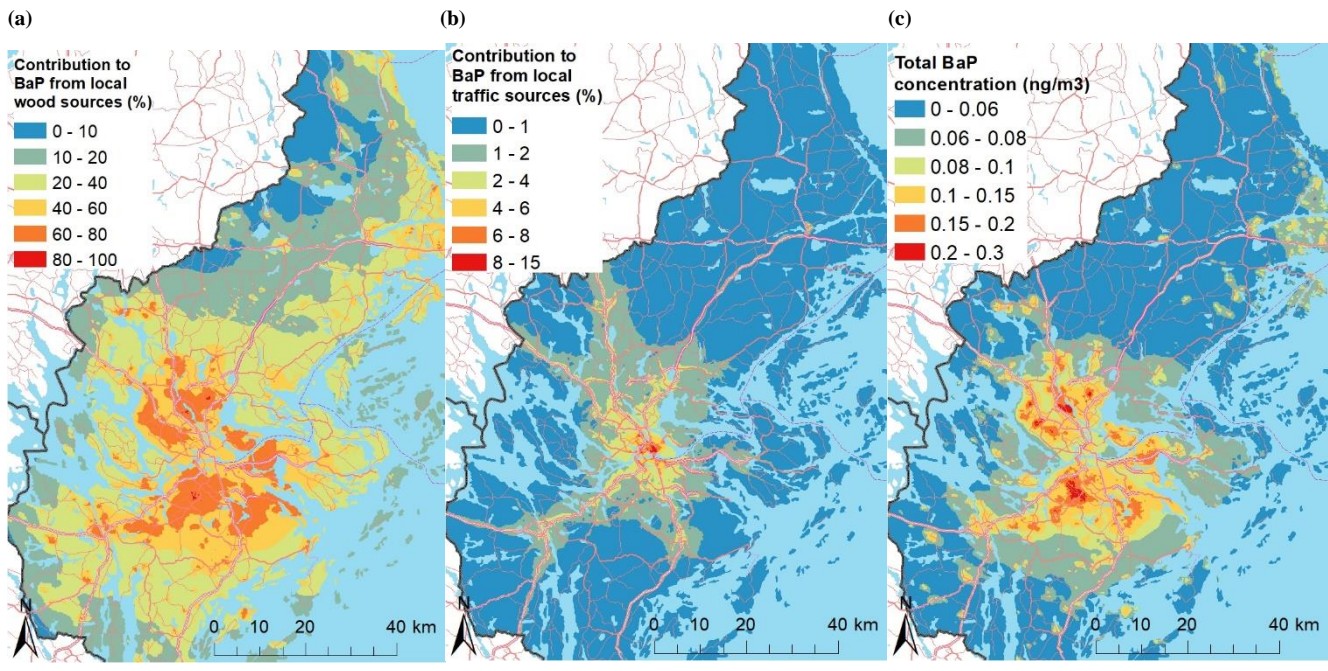

**Figure 10**. Modelled contribution to B[*a*]P from **(a)** local residential wood burning and **(b)** traffic in Stockholm County for a normal meteorological year and **(c)** modelled total B[*a*]P concentration in Stockholm County for a normal meteorological year with regional background data from 2014-2018. Red lines show road net and dark grey lines show the border of the County.

**4.6 B[*a*]P equivalency concentration**

The annual mean B[*a*]P$_{eq}$ concentration from all three locations was 1.03 ng m$^{-3}$ with a large summer-to-winter variation (0.09-2.15 ng m$^{-3}$). As shown in Table S6, the low-trafficked residential area (DE) exhibited the most dramatic variation in B[*a*]P$_{eq}$ levels throughout the year, ranging from 0.06 to 3.56 ng m$^{-3}$, followed by the two urban areas (EN: 0.08- 2.82 ng m$^{-3}$, YJ 0.07-2.76 ng m$^{-3}$). The levels were comparable to those reported from the residential area in Dettenhausen (Germany), 2.7 and 3.4



ng m$^{-3}$ during the winters 2005/2006 and 2009, respectively (Bari et al., 2011). At an urban background site in Thessaloniki (Greece) slightly lower values were observed during the cold season, 2.0 and 1.4 ng m$^{-3}$ (Sarigiannis et al., 2015; Manoli et al., 2016). The Greek studies used smaller RPF values, initially proposed by Nisbet and LaGoy as toxic equivalency factors (TEFs) (Nisbet and LaGoy, 1992), than those used in this study, resulting in slightly underestimated B[$a$]P$_{eq}$. Later studies

using updated TEFs (Muller, 1997; Boström et al., 2002) showed higher concentrations than those from the present study. As an example, 4.0 ng m$^{-3}$ was measured at the urban site in Oporto (Portugal) during the winter (Slezakova et al., 2013). Another study from six sampling stations in north-west Italy reported an annual mean B[$a$]P$_{eq}$ of 2.7 ng m$^{-3}$ (Khan et al., 2018), probably underestimated due to the use of smaller factors. The B[$a$]P$_{eq}$ at the urban background (TK) in the present study was 0.43 ng m$^{-3}$ (0.15-1.06 ng m$^{-3}$), similar to the spring average B[$a$]P$_{eq}$ (0.52 ng m$^{-3}$) obtained from the other three locations. It is also

noted that the B[$a$]P concentration constituted approximately 10% of the summed B[$a$]P$_{eq}$, 10.3 and 11.2% in the urban background and residential areas, respectively. In addition, the PAH profiles are compared after TEF adjustment from three locations as discussed in S3. Those PAHs with higher TEF values, e.g. B[$c$]f, B[$j$]A, DB[$a,h$]A, DB[$a,l$]P, become predominant after the conversion, despite their lower abundances as shown in Fig. S3 in S2.

**4.7 Population exposure and calculated cancer cases**

Cancer risk assessment have been evaluated for the measured concentrations of PAHs within this work. We used a previous method published in 2017 (Dreij et al., 2017). The method is based on *in vitro* data in which potency factors of several PAHs are assessed. The resulting cancer risk is significantly higher than that from the established method using separate *in vivo* risk factors. In short we have used 100 × 100 m gridded population residential data for Stockholm County together with the

calculated concentrations of B[$a$]P from local wood burning sources and local traffic sources, based on normal meteorology, e.g. climatology. The regional background is assumed to be equal throughout the area and is a five-year average (2014-2018) from the regional background station in Råö. The result is population weighted concentrations of B[$a$]P. The cancer risk is finally assessed using the weighted B[$a$]P concentration together with 12 PAH profiles for each source, i.e. traffic, wood, and the regional background. The assumed relative abundances are shown in Table 3 together with the RPF values. The mixture

potency factors (MPFs) used are 487 and 1094, which were derived from analyses of filter samples in Stockholm, in a car tunnel and at a suburban area (Dreij et al., 2017). The result from population exposure risk estimations is shown in Table 4. On an average, 168 cancer cases or 7.2 cases per 1 million inhabitants, are estimated to be caused yearly by exposure to PAHs in Stockholm County. According to the National Board of Health and Welfare (Socialstyrelsen) in Sweden there were 3432 lung related cancer cases during 2014-2017. This means that approx. 20 % of all lung cancer cases in Stockholm County could

be caused by airborne PAHs when applying MPFs, although a percentage of 6.4 has been reported previously (Dreij et al., 2017). However, it should be noted that using traditional individual potency factors result in far less cancer cases (< 1) per year.

Table 3. PAH profiles assumed for the various sources and used RPF.


| PAH | Traffic (%) | Wood (%) | Regional background (%) | RPF (Dreij et al 2017) | |
|---|---|---|---|---|---|
| B[a]P | 17.25 | 13.82 | 12.56 | 1 | |
| DB[a,l]P | 0.49 | 0.21 | 0.18 | 30 | |
| B[b]F | 17.66 | 23.76 | 22.92 | 0.8 | |
| B[k]F | 6.62 | 11.37 | 9.26 | 0.03 | 5 |
| DB[a,h]A | 0.69 | 1.89 | 2.34 | 10 | |
| I[1,2,3-cd]P | 7.81 | 10.39 | 12.48 | 0.07 | |
| DB[a,h]P | 0.08 | 0.06 | 0.18 | 0.9 | |
| DB[a,i]P | 0.17 | 0.30 | 0.59 | 0.6 | |
| DB[a,e]P | 1.04 | 1.41 | 2.08 | 0.4 | |
| B[e]P | 19.32 | 17.11 | 14.48 | 0.002 | |
| Cor | 9.00 | 5.53 | 7.73 | 0.01 | 10 |
| B[ghi]p | 19.88 | 14.15 | 15.19 | 0.009 | |

See Table S1 for compound abbreviations.

Table 4. Estimated lung related cancer cases in Stockholm County.

| | | Estimated cancer cases (%) |
|---|---|---|
| Total Stockholm County | Local wood | 48 (29%) |
| | Local traffic | 4 (2%) |
| | Regional background | 116 (69%) |
| | Total | 217   168 (100%) |
| 0.1 million Stockholm inhabitants | Local wood | 2.0 (29%) |
| | Local traffic | 0.2 (2%) |
| | Regional background | 5.0 (69%) |
| | Total | 7.2 (100%) |

## 5 Summary and conclusions

In this study the measurement of PAHs and sugars in air $PM_{10}$ was done from one urban background and three residential areas in Sweden to understand the influence of wood burning on PAHs. A clear seasonal variation of PAHs was observed, especially during colder season due to the increased residential heating and meteorology. The seasonal shift in the distribution of low and high molecular weight PAH between gas and particle phase was present. In addition, the increased sugar level reassured the





increased wood burning during colder season. The sugar ratio, levoglucosan/(mannosan+galactosan), is a useful marker for identifying the wood burning source in the three residential areas. The estimated burning source was consistent with the observations in each area such as the preferences of the residents for wood type, garden waste burning, domestic wood burning etc. Periods with intense wood burning was denoted by elevated sugar ratios.

The measurement during the Walpurgis Night showed high sugar emissions due to the increased wood burning. The event resulted in elevated levoglucosan, B[a]P and OPAH concentrations. Interestingly, OPAH levels increased as much as for levoglucosan, while the B[a]P level increased to a lesser extent. This suggests that the level of total OPAHs as combined with levoglucosan could be a good indicator for wood burning. Both B[a]P and levoglucosan levels highly correlated with PM$_{2.5}$ concentrations, although levoglucosan is strongly correlated with black carbon and OPAHs. Since this study measured OPAHs

in air PM$_{10}$ in the urban background, the levoglucosan-OPAH relationship could be further investigated in relation to the measurement at residential areas.

Either the levoglucosan tracer method or the model did not fully explain the measured B[a]P concentrations. Especially for the cold season, showing underestimated B[a]P levels in the levoglucosan tracer method whilst the modelled data based on the emission inventory consistently overestimated B[a]P. The model indicates that the local wood sources contribute to 98 % of

all B[a]P emissions within Stockholm County and the local traffic to the remaining 2 %. At the residential area measurement sites in Stockholm, approximately 30 % of the B[a]P is non-local according to calculations from an emission inventory within this present work. From local emissions, wood sources contribute to 60-70 % which leaves up to 2 % from traffic sources. In the urban background area traffic sources are calculated to contribute 1/8 of the B[a]P concentration from local sources and wood the remaining 7/8 of the B[a]P. In urban background non-local sources are dominating as source of B[a]P contributing

to around 60 %.

The B[a]P equivalency concentration shows a high winter-to-summer ratio, posing an underestimated risk on a yearly basis. The lung cancer related risk assessment estimates 13.4 cancer cases per 0.1 million inhabitants in Stockholm County, caused by airborne particulate PAHs, accounting for almost 40 % of the total lung cancer prevalence in the region when assuming that the modelled concentrations of B[a]P is valid. Since modelled data is shown to be higher than measured values, we can

conclude that there is a significant uncertainty in the prediction in cancer cases, thus highlighting the importance of air quality measurements.

## Appendix A

### Input data for emission inventory of B[a]P from residential wood combustion in Stockholm County

The emission inventory of residential wood combustion includes approximately 0.11 million objects based on chimney

sweeper documentations in 7 municipalities in Stockholm county obtained from Greater Stockholm fire brigade. In the other parts of the county standardized sources have been placed among residential area, based on housing data from Statistics Sweden (Statistiska centralbyrån, or SCB) with the assumption that the share of stoves, boiler and fire places etc. are equal in the region





as in the available sweeper data. Except for wood combustion sources based on chimney sweeper data there are approximately 0.17 million standardized sources in Stockholm County. There are various parameters included in the estimated amount of B[*a*]P emission are described in sections below.

*Combustion system*

From the chimney sweeper data there are 110 348 combustion units aggregated into four main categories according to Table A1 below.

Table A1. Share of each combustion system type from chimney sweeper data in the municipalities of Danderyd, Stockholm,
Täby, Värmdö, Vaxholm, Vallentuna and Österåker.

| Combustion system | Share of objects (%) |
|---|---|
| Wood boiler | 14 |
| Pellets/ wood chip boiler | 1 |
| Open fire place | 81 |
| Oil /gas boiler | 4 |

*Energy consumption*

The annual energy consumption has previously been assessed by the Swedish Meteorological and Hydrological Institute (SMHI) with the ENLOSS model. (Taesler and Andersson, 1984; Taesler et al., 2006)) The size of housing and the weather
affects the required need for heating. A general energy consumption has been assumed based on an average residential housing with 152 m$^{-2}$ living space in Stockholm County. That assumption is made for a normal meteorological year during the reference period 1960-1990. The needed annual energy need is estimated to be 13 259 kW h house within the whole county.

*Low emission wood boiler*

There are environmentally certified boilers and regular boilers in use. The emission from the types vary vastly and therefore it is of importance to estimate how many boilers of each type that is used. According to a national inventory 695 000 local heating systems with 230 000 wood boilers the low emission type accounted for approximately 30 % of the total from wood boilers (Todorovic et al., 2007). In a more recent survey by SMHI during 2018 in areas outside Stockholm County (Skellefteå, Strömsund, Alingsås) the results were approximately the same, 27 % of the wood boilers were low emission types (Andersson
et al., 2018). In the emission inventory for Stockholm, the amount of low emission boilers was assumed to be 30 %.

*Efficiency*

The efficiency for the type of boilers or fireplace is shown in Table A2 and were previously reported by SMHI (Omstedt et al., 2014; Andersson et al., 2015).



Table A2. Efficiency for boilers and fireplaces used in emission inventory (Omstedt et al., 2014; Andersson et al., 2015).

| Combustion system | Efficiency (%) |
|---|---|
| Wood boiler | 60 |
| Low emission wood boiler | 75 |
| Pellets / wood chip boiler | 75 |
| Open fire place | 70 |
| Oil / gas boiler | 90 |
| Averaged boiler [a] | 70 |

[a] Used where no chimney sweeper data is available

*Utilization*

5 The utilization of the boiler depends both on the boiler type and also whether the building can use the district heating system. The utilization assumed in the emission inventory for Stockholm is based on two previous firing habit surveys within Sweden from SMHI and can be read in Table A3 (Omstedt et al., 2014; Andersson et al., 2018). Note that only boilers are affected by whether the house is within the district heating system. Open fireplaces are commonly used for leisure firing rather than heating.

Table A3. Utilization depending on boiler and district heating.

| Combustion system | District heating area: Utilization (%) | Outside district heating area: Utilization (%) |
|---|---|---|
| Wood boiler | 21 | 63 |
| Pellet/wood chip boiler | 61 | 61 |
| Open fire place | 11 | 11 |
| Oil/gas boiler | 100 | 100 |

*Emission factors*

All the previously mentioned factors have been used to estimate the emission for each boiler or fireplace in the emission
15 inventory. An average emission factor has been calculated based on the known chimney sweeper data for the areas in municipalities where no chimney sweeper data was available. It is known that the individual habits are important for the amount of emission. The moisture in the wood and the oxygen supply during combustion has a substantial effect on the emissions. However, the used emission factored listed in Table A4 were based emission data in various previous work and can be considered as a base case (Todorovic et al., 2007; SMED and Naturvårdsverket, 2017; EMEP/EEA, 2013).
20 An assumption that have been made is that 10 % of the wood boilers have bad combustion, e.g. moist wood or smoldering fire.

Table A4. B[a]P emission factors used in emission inventory in Stockholm County.



| Combustion system | B[a]P (mg MJ$^{-1}$) | **Low emission factor** | **High emission factor** |
|---|---|---|---|
| Wood boiler | 0.12 | 0.09 | 0.38 |
| Low emission wood boiler | 0.02 | 0.001 | 0.09 |
| Weighted average wood boiler | 0.09 | 0.063 | 0.29 |
| Pellet/wood chip boiler | 0.01 | 0.00001 | 0.12 |
| Wood stove, fire place etc | 0.05 | 0.004 | 0.27 |
| Oil/gas boiler | 0.001 | 0.001 | 0.001 |
| Averaged boiler [a] | 0.054 | **0.012** | **0.26** |

[a] Used where no chimney sweeper data is available

*Chimney*

In the database each chimney is assumed to have the same properties regarding proportions and gas flow etc. All point sources properties are listed in bullet below.

- Height: 5 m
- Gas temperature: 100 ℃
- Gas flow: 1 m s$^{-1}$
- Outer / inner diameter of chimney: 0.8 / 0.5 m

*Emission variation*

Since the meteorology changes during the year and hourly assumptions were also made about the emission variation. The assumptions depend on the boiler or fireplace and were based on a master's project written in 2000 in which a survey was one to determine heating habits in the Swedish town Vännäs (Andersson, 2000).

Table A5. Share of total yearly combustion during each month used in emission inventory.

| Month | Share from total combustion (%) |
|---|---|
| January | 15 |
| February | 12 |
| March | 12 |
| April | 10 |
| May | 6 |
| June | 3 |
| July | 2 |
| August | 3 |
| September | 4 |



| | |
|---|---|
| Oktober | 9 |
| November | 11 |
| December | 13 |

Table A6. Share of total daily combustion used in emission inventory for open fireplace, oil/gas boiler and pellet/wood chip boiler.

| Hour (start) | Relative combustion for open fireplace (%) | Relative combustion for oil/gas boiler (%) | Relative combustion for pellet/wood chip boiler (%) |
|---|---|---|---|
| 00 | 0 | 4 | 0 |
| 01 | 0 | 4 | 0 |
| 02 | 0 | 4 | 0 |
| 03 | 0 | 4 | 0 |
| 04 | 0 | 4 | 0 |
| 05 | 0 | 4 | 8 |
| 06 | 0 | 4 | 8 |
| 07 | 0 | 4 | 8 |
| 08 | 0 | 4 | 8 |
| 09 | 0 | 4 | 0 |
| 10 | 0 | 4 | 0 |
| 11 | 0 | 4 | 0 |
| 12 | 0 | 4 | 0 |
| 13 | 0 | 4 | 0 |
| 14 | 3 | 4 | 0 |
| 15 | 3 | 4 | 0 |
| 16 | 3 | 4 | 0 |
| 17 | 22 | 4 | 17 |
| 18 | 22 | 4 | 17 |
| 19 | 22 | 4 | 17 |
| 20 | 22 | 4 | 17 |
| 21 | 0 | 4 | 0 |
| 22 | 0 | 4 | 0 |
| 23 | 0 | 4 | 0 |

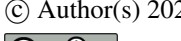



Table A7. Share of total daily combustion used in emission inventory for wood boilers separated in weekdays and Saturday, Sunday.

| Hour (start) | Mon-Fri | Sat | Sun |
| --- | --- | --- | --- |
| 00 | 0 | 0 | 0 |
| 01 | 0 | 0 | 0 |
| 02 | 0 | 0 | 0 |
| 03 | 0 | 0 | 0 |
| 04 | 0 | 0 | 0 |
| 05 | 9 | 9 | 9 |
| 06 | 9 | 9 | 9 |
| 07 | 9 | 9 | 9 |
| 08 | 9 | 9 | 9 |
| 09 | 2 | 2 | 2 |
| 10 | 2 | 2 | 2 |
| 11 | 2 | 2 | 2 |
| 12 | 3 | 3 | 4 |
| 13 | 3 | 3 | 4 |
| 14 | 3 | 3 | 3 |
| 15 | 0 | 0 | 0 |
| 16 | 3 | 3 | 3 |
| 17 | 3 | 3 | 3 |
| 18 | 10 | 10 | 10 |
| 19 | 10 | 10 | 10 |
| 20 | 10 | 10 | 10 |
| 21 | 10 | 10 | 10 |
| 22 | 0 | 0 | 0 |
| 23 | 0 | 0 | 0 |




**Supplement** The supplement related to this article is available online at: https://doi.org/xxxxxxxxxxxxxxxxxxxxxxxxxxxx.

**Author contribution** S. Silvergren handled and collected samples, created emission data and performed dispersion modelling, H. Lim, S. Spinicci, and F.M. Rad performed PAH, OPAH and sugar analysis, and H. Lim prepared the manuscript with contributions from all co-authors.

**Competing interests** The authors declare that they have no conflict of interest.

**Acknowledgements** This project was funded by the Stockholm County Council (Stockholms läns landsting), Stockholm Environment and Health Administration and Stockholm University.

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
