# Peer review of "Contribution of wood burning to exposures of PAHs and oxy-PAHs in Eastern Sweden"

_Atmospheric Chemistry and Physics, 2022_

## Author Comment (AC1)

We thank the referee for the time and effort to review the manuscript and appreciate the insightful thoughts and comments. Below we respond to all the concerns and suggestions, and the manuscript will be revised accordingly.

We have highlighted our responses in blue with the referee's original comments in black.

RC1: 'Comment on acp-2022-20', Anonymous Referee #1, 04 May 2022

1. In the study concentration levels of PAHs, sugars and OPAHs in PM10 were studied at one urban background and three residential areas in Sweden and results were compared to the modelling results. Population exposure was also estimated, which was interesting and showed the impact of the study more clearly. Sampling and analytical methods were well-described and manuscript is well-written. There have been some earlier publications on PAHs and wood combustion from Nordic countries, but there is very little knowledge on the OPAHs. Therefore results on OPAHs could be discussed more. In addition to this, I had only minor comments:

Response: We agree with the referee that the OPAH results could be further discussed to better understand atmospheric OPAHs. We greatly appreciate the referee's comment and will revise "Section 3.3" accordingly as below.

3.3 Impact of non-residential biomass burning and OPAHs

The measured levels of PAHs, OPAHs and sugars in airborne $PM_{10}$ taken at the urban background (TK) around the Walpurgis Night (Apr 30) in 2016 are shown in Fig. 6. A clear elevation of both the total sugar and levoglucosan can be observed in the sample collected on the event day. In the present study, the total sugar level showed an increase of almost three times, from 36.4 to 117 ng m$^{-3}$. The total OPAH level was also shown to increase approximately threefold. The sugar ratios (4.4-7.2) from the urban background indicated a mixed emission source of hard and soft wood combustion, except for the sample collected on the 30$^{th}$ Apr. This sample exhibited sugar ratio of 8.8, i.e. reflecting hard wood burning. Detailed information is given in Table S5.

The dominating OPAH was 7$H$-benzo[$de$]anthracen-7-one (benzanthrone, BAQ), accounting for more than 50% of the measured OPAHs (55 - 76%), followed by 9,10-anthracenedione (AQ) (4 - 28%) (Fig. xx). The same trend has been reported in atmospheric $PM_{2.5}$, measured at an urban background site in Bologna, Italy (Pietrogrande et al., 2022). The correlation between OPAH and its parent PAH was significant ($p$ < 0.01), indicating PAHs and OPAHs were from the same primary combustion source or PAHs were the secondary source of OPAHs. However, the increased emission of OPAHs on the Walpurgis Night is mainly from the primary combustion source, i.e. biomass burning (Fig. xx).

Pietrogrande, M.C., Bacco, D., Demaria, G., Russo., M, Scotto., F. and Trentini, A: Polycyclic aromatic hydrocarbons and their oxygenated derivatives in urban aerosol: levels, chemical profiles, and contribution to $PM_{2.5}$ oxidative potential, Environ. Sci. Pollut. Res., doi.org/10.1007/s11356-021-16858-z, 2022.

[Figure]

Figure xx. Measured concentration of total OPAHs (AQ, CPPQ, BAQ, and B[*a*]AQ) and individual concentration of OPAH and its parent PAH in PM$_{10}$ collected from Stockholm urban background (TK) during Feb 22 – May 5, 2016.

2. Page 2, line 30: Lifetime of levoglucosan has been estimated to be 1.8 days on average. So it is not that long living (Li, Y., Fu, T-M., Yu, J.Z., Feng, X., Zhang, L., Chen, J., Suresh Kumar Reddy, B., Kawamura, K., Fu, P., Yang, X., Zhu, L., and Zeng, Z.: Impacts of Chemical Degradation on the Global Budget of Atmospheric Levoglucosan and Its Use As a Biomass Burning Tracer, Environmental Science & Technology,55, 5525-5536, https://doi.org/10.1021/acs.est.0c07313, 2021).

Response: We thank the referee that there are other studies indicating the atmospheric degradation of levoglucosan during a short period than previously reported. We greatly appreciate the referee's comment and will revise the text accordingly as below.

"A highly selective tracer for burning of wood is levoglucosan (1,6-anhydro-β-D-glucopyranose), a monosaccharide derivative formed when cellulose is pyrolysed at high temperatures (Shafizadeh, 1968; Simoneit et al., 1999). Due to the high concentration in the smoke and a high chemical stability, this tracer compound can be detected in the atmosphere through long-range transports (Simoneit et al., 1999; Fraser and Lakshmanan, 2000). In addition, mannosan and galactosan released from the thermal degradation of hemicellulose are also detected in wood smoke emissions and suggested to be source-specific tracers for wood burning (Nolte et al., 2001; Simoneit, 2002). However, recent studies indicate that atmospheric degradation of levoglucosan is important and may shorten the lifetime significantly (Li et al., 2021). They suggest considering the aging in air masses of levoglucosan when it is used to calculate the biomass burning contribution to organic

carbon (Hong et al., 2022). In our case the sites were located in close proximity to the main sources of levoglucosan and the photochemical degradation is expected to be insignificant for these conditions."

Li, Y., Fu, T.-M., Yu, J.Z., Feng, X., Zhang, L., Chen, J., B, S.K.R., Kawamura, K., Fu, P., Yang, X., Zhu, L. and Zeng, Z.: Impacts of chemical degradation on the global budget of atmospheric levoglucosan and its use as a biomass burning tracer, Environ. Sci. Technol., 55 (8), 5525-5536, doi: 10.1021/acs.est.0c07313, 2021.

Hong, Y., Cao, F., Fan, M.-Y., Lin, Y.-C., Gul, C., Yu, M., Wu, X., Zhai, X. and Zhang, Y.-L.: Impacts of chemical degradation of levoglucosan on quantifying biomass burning contribution to carbonaceous aerosols: A case study in Northeast China, Sci. Total Environ., 819, 152007, doi.org/10.1016/j.scitotenv.2021.152007, 2022.

3. Page 3, lines 1-3: Where? In this study or earlier studies?

Response: We understand that it was not clearly written in the manuscript where they were measured. It referred to previous studies measured in various cities of central and northern European countries. We greatly appreciate the referee's comment and will revise the text accordingly as below.

"Earlier studies have measured either levoglucosan, or the combination of all three monosaccharides, to estimate the contribution of wood burning to the total air PM collected in various cities of central and northern European countries (Yttri et al., 2005; Caseiro et al., 2009; Maenhaut et al., 2012, 2016; Wagener et al., 2012; Glasius et al., 2018)."

4. Maybe you could mention somewhere that some of the PAHs you studied are semivolatile and significant fraction of them may be found in gas phase.

Response: We agree with the referee, and this will be added in the section "3.1 Seasonal variation of PAH and sugar levels. PAHs". The revised text will be as below.

"The PAH concentrations showed a strong seasonal variation at all three sites as shown in Fig. 3. On the left side, the mass concentration of low and high molecular weight PAHs (LMW and HMW) are compared together with those of the total PAHs. 11 PAHs with three and four rings were grouped in LMW while 22 five- and six-ring PAHs were in HMW. PAHs are semi-volatile compounds and partitioned in gas and particle phases depending on the volatility. Three- and four-ring PAHs are distributed mostly in the gas phase during warmer season and *vice versa* (Bi et al., 2003). The same applied to this study where LMW PAH levels were lower during summer due to their partitioning more in the gas phase. The pie chart (right side in Fig. 3) shows the relative abundance of LMW and HMW during summer (Jun-Jul) and winter (Jan-Feb). A considerable shift in the relative LMW level from winter to summer was observed in all three locations with the largest change in DE. In addition, the distinctive seasonal shift observed in DE was from the increased residential heating, which mostly affected the PAHs with four rings. The increased emission of four-ring PAHs from domestic

heating was also reported in the high Arctic during winter (Singh et al., 2017). The sugar levels, in general, followed the seasonal variation as the PAHs, however, there were occasions with increased sugar levels when biomass burning or wood combustion happened."

Bi, X., Sheng, G., Peng, P., Chen, Y., Zhang, Z., and Fu, J.: Distribution of particulate- and vapor-phase n-alkanes and polycyclic aromatic hydrocarbons in urban atmosphere of Guangzhou, China, Atmos. Environ., 37 (2), 289-298, doi:10.1016/S1352-2310(02)00832-4, 2003.

5. Could you also comment somewhere how close to the EU annual limit values/thresholds you were?

Response: We agree with the referee that it could be mentioned how the measurement in this study was in relation to the EU annual limit. The comparison between the measured and EU target annual mean B[$a$]P concentrations are made and will be inserted in the section "3.2 PAH and sugar emission trend associated with residential wood burning" as below.

"The annual mean concentration of B[$a$]P from the three sites (DE, EN, YJ) and one rural background (ASP) were 0.11, 0.08, 0.09, and 0.06 ng m$^{-3}$, respectively (Table S4), which is far below than the EU target value of 1 ng m$^{-3}$."

6. Page 8, lines 22-24: Could it also be that during summer time more of these 3 and 4 ring PAHs partition to the gas phase?

Response: We agree with the referee, and this will be added in the section "3.1 Seasonal variation of PAH and sugar levels. PAHs". Please see the revised text above (Response to comment # 4).

7. Figure 4: Which are the locations if fig a)-c)?

Response: We appreciate the referee's comment that the information of the location is missing in Fig. 4. The figure legend will be revised as below.

"Figure 4. Distribution of annual measurement data for levoglucosan, mannosan, and galactosan from three sampling locations. Mannosan and galactosan are determined as sum due to co-elution in the chromatogram: **(a)** DE, **(b)** EN and **(c)** YJ."

8. Figure 8: For EN and YJ local wood and traffic has same pattern in the figure.

Response: We appreciate the referee's comment and will change the data format to differentiate four data series as below.

[Figure]

---

## Author Comment (AC2)

Response to RC2: 'Comment on acp-2022-20', Anonymous Referee #2, 05 May 2022 See author response in blue as below.

The manuscript is well written and exposes results from the air monitoring campaign. In my opinion, the manuscript must be focused on that line of research and could be considered after major revisions.

**Major comments:**

1. The aim of the study is related to showing the results of the air monitoring campaign meanwhile the authors tried to include numerical simulation in their discussion-analysis with a small contribution to the analysis of the measurements during the campaign. The authors used a Gaussian Model without considering the photochemical reactions of PAHs in the air, even when they mentioned the formation of the OPAHS from the photochemical degradation of PAHs (Line26 p2). In my opinion, the authors must consider a more confident model (like Eulerians) to replicate the observations.

Response: The photochemical degradation of particle phase PAH is a very slow process – for most particle bound compounds it takes several hours or even days (Keyte et al., 2013\*). During winter at high latitudes, with relatively little solar radiation and low temperatures, photochemical oxidation processes will be less important. This is true also for the formation of OPAHs from their parent-PAHs. Since transport time from local emissions to the sampling sites is very short (minutes) in comparison to photochemistry we consider the particle bound PAH to be inert, which justifies using a Gaussian model without photochemistry. This means that we can safely assume that the OPAH concentrations are from direct emissions rather than photochemical formation of locally emitted PAH. Some OPAHs are long-range transported to our site, and they may have been formed in photochemical reactions, but we only include local emissions in the dispersion modelling – long-range transport is added to the modelled concentrations based on background measurements.

We also want to stress that the Gaussian air quality dispersion model used here has been described and used successfully in several peer reviewed studies, both epidemiological and health impact assessments (examples listed below). Both in this study and in many other, it is not the dispersion model but the emission input that are most critical when it comes to uncertainties in the modelling. Especially in the case of wood burning (see also below).

\*Keyte, I.J., Harrison, R.M. and Lammel, G: Chemical reactivity and long-range transport potential of polycyclic aromatic hydrocarbons – a review, Chem. Soc. Rev., 42, 9333 - 9391, https://doi: 10.1039/c3cs60147a, 2013.

Examples of publications using the Gaussian AQ dispersion over Greater Stockholm:

Dreij, K., et al. Cancer Risk Assessment of Airborne PAHs Based on in Vitro Mixture Potency Factors. Environmental Science & Technology 2017 51 (15), 8805-8814. DOI: 10.1021/acs.est.7b02963.

Johansson, C.; Burman, L.; Forsberg, B. The effects of congestions tax on air quality and health. Atmos. Environ. 2009, 43, 4843–4854.

Kriit, H., Nilsson Sommar, J., Forsberg, B., Åström, S., Svensson, M., Johansson, C. A health economic assessment of air pollution effects under climate neutral vehicle fleet scenarios in Stockholm, Sweden. Journal of Transport & Health, 22, 101084, 2021.

Ljungman, P.L.S.; Andersson, N.; Stockfelt, L.; Andersson, E.M.; Nilsson Sommar, J.; Eneroth, K.; Gidhagen, L.; Johansson, C.; Lager, A.; Leander, K.; Molnar, P.; Pedersen, N.L.; Rizzuto, D.; Rosengren, A.; Segersson, D.; Wennberg, P.; Barregard, L.; Forsberg, B.; Sallsten, G.; Bellander, T.; Pershagen, G. Long-term exposure to particulate air pollution, black carbon and their source components in relation to ischemic heart disease and stroke, Environ. Health Perspect. 127 (10), 2019. https://ehp.niehs.nih.gov/doi/10.1289/EHP4757

Segersson, D.; Eneroth, K.; Gidhagen, L.; Johansson, C.; Omstedt, G.; Nylén A.E.; Forsberg, B. 2017. Health Impact of PM10, PM2.5 and Black Carbon Exposure Due to Different Source Sectors in Stockholm, Gothenburg and Umea, Sweden. Int J Environ Res Public Health | 14 (7), 2017. https://www.mdpi.com/1660-4601/14/7/742

Segersson, D.; Johansson, C.; Forsberg, B. 2021. Near-Source Risk Functions for Particulate Matter Are Critical When Assessing the Health Benefits of Local Abatement Strategies. Int J Environ Res Public Health, 18, 2021. https://www.mdpi.com/1660-4601/18/13/6847

Sommar et al., 2021. Long-term exposure to particulate air pollution and black carbon in relation to natural and cause-specific mortality: a multicohort study in Sweden. BMJ open, 11, 2021. 10.1136/bmjopen-2020-046040. https://bmjopen.bmj.com/content/11/9/e046040

Wu et al., 2022. Air pollution as a risk factor for Cognitive Impairment no Dementia (CIND) and its progression to dementia: A longitudinal study. Environment International, 160, February 2022, 107067.

In Section "2.4.4 Model simulations", the text will be revised to justify the use of Gaussian model as below.

"All dispersion modelling of local emissions was performed with a Gaussian plume model in the Airviro air quality management system (Apertum, 2021; Segersson et al., 2017). The photochemical degradation of particle phase PAH is a very slow process (Keyte et al., 2013\*) and the local emission source was very close to the sampling sites in the study. Therefore, a Gaussian model without photochemistry can be used, assuming that the OPAH concentrations are from direct emissions rather than photochemical formation of locally emitted PAH.

2. P7 L15 Could be more detailed about how the climatology data was used and applied in the study (ie: the data was averaged? are measured data in specific places? how many places?) How the historical data could fit the simulation in the period of analysis? The authors must justify this method to reduce the uncertainties about meteorological conditions for the PM dispersion.

Response: We use detailed data from a meteorological mast, in addition to the regular data (wind speed, wind direction, temperature, relative humidity, solar radiation), we have standard deviation of vertical wind speed and differential temperature to estimate the stability and mixed layer height. Hourly meteorological data was used to model the local contribution of B[a]P at the measurement sites. Using modeled local contributions with added measured monthly regional background from the same month we derived total B[a]P concentrations that was compared to measured values in Stockholm County. However, the climatology was used to model local B[a]P in the entire Stockholm County and it should also be noted that we added the average regional background during 2014-2018 to obtain total concentrations. This procedure was chosen in order to assess a general exposure and health impact in which we do not want to introduce a bias due to the conditions during our measurement campaign.

The climatology consists of frequency of occurrence of hourly mean data in 60 wind sectors, each 6°, with 6 stability classes. All events falling into a specific sector will be classified according to the atmospheric

stability conditions (as default discriminated by 6 intervals of Monin-Obukhov lengths). When all data have been sorted, frequencies of all classes will be estimated and the median values (of the Monin-Obukhov lengths) of each class (in this case 360 classes) are identified, including the specific date/hour when each class example occurred.

In order to extrapolate over the whole model domain, we use the diagnostic wind model that takes into account topography, roughness and land-use.

In Segersson et al (2017) we demonstrate the consistency in averages estimated using the two methods by comparing with hourly mean time series calculations for 7000 receptor points in the same domain as in this study. The comparison shows that the results are very similar; slope =  $0.891 \pm 0.002$ ; intercept =  $0.071 \pm 0.021$ ; correlation, r = 0.99, indicating that the climatology based calculations well represents the full hourly time-series.

In Section "2.4.4 Model simulations", the text will be revised accordingly:

"Airviro uses a diagnostic wind model based on the theory first described by Danard (Danard, 1977), in which it is assumed that small scale winds can be seen as a local adaptation of large-scale winds. Calculation of B[a]P concentrations was performed for local wood and traffic sources in a 5.5 km2 size area with a 100 m grid size around the measurement sites using meteorology during the measurement period from a 50 m meteorological mast located in Högdalen, a suburb in Stockholm. Meteorological data includes the wind speed, wind direction, temperature, relative humidity, and solar radiation. Modelled monthly mean concentrations due to local residential wood combustion and road traffic at the Stockholm county measurement sites was based on hourly time series calculations using hourly meteorological from the Högdalen mast. The measured monthly concentration of B[a]P in regional background was added to the calculated local amount and the total B[a]P concentration can then be compared with the measured concentration during the measurement campaign in Stockholm urban background (TK), residential area (EN) and (YJ). The measurement in the northern residential area (DE) is outside of the emission inventory area and cannot be compared with calculated concentrations.

Dispersion modelling was also done for entire Stockholm County for local wood and local traffic sources using climatology for 1993-2010. The climatology data consists of frequency of occurrence of hourly mean data in 60 wind sectors, each 6°, with 6 stability classes, which are classified according to the atmospheric stability conditions (as default discriminated by 6 intervals of Monin-Obukhov lengths). After data sorting, frequencies of all classes is estimated and the median values (of the Monin-Obukhov lengths) of each class (in this case 360 classes) are identified, including the specific date/hour when each class example occurred. We use the diagnostic wind model that takes into account topography, roughness and land-use in order to extrapolate over the whole model domain. The climatology derived modelled data was used to calculate yearly B[a]P concentrations in entire Stockholm County in 100x100 m grid size in order to obtain a general exposure and health impact assessment for average meteorological conditions. The average regional background B[a]P concentrations during the years 2014-2018 was added to the modelled local contribution. "

3. P13 L15 The authors showed the causes for the poor correlation between modeled and observed values of PAHs. This result does not contribute to the analysis. The numerical simulation could probably be erased from the manuscript due to the several causes mentioned by the authors and the limitation of the model used.

Response: We appreciate the referee's comment, but as we point out that the model calculations of concentrations due to emissions from residential wood combustion is very uncertain due to the many uncontrollable factors associated with wood burning, like wood characteristics (wetness, type of wood...), burning efficiency that depends not only on types of stoves but also on individual wood users' behavior

etc. Even if there are these uncertainties, dispersion modelling is necessary in order to estimate spatial variations and population exposures, which we believe adds important information to the paper (also pointed out by referee no 1.).

Minor comments:

4. p4 subsection 2.2 Why only used the PM10 filter and not PM2.5? The PM2.5 sampling filter could indicate more confidence in the long-range transportation and source contribution of the PAHs and OPAHs.

Response: We do not think using  $PM_{2.5}$  would make it possible to distinguish long-range transport (versus locally emitted) PAH and OPAH since most (if not all) PAH and OPAH are in the submicrometer size fractions REFs, which means that the concentrations of PAH and OPAH will be the same in  $PM_{10}$  and  $PM_{2.5}$ . But the argument for using  $PM_{10}$  was that the current air quality legislation of PAHs is based on  $PM_{10}$  (EU 2005).

EU: Council Directive 2004/107/EC of the European Parliament and of the Council of 15 December 2004 relating to arsenic, cadmium, mercury, nickel and polycyclic aromatic hydrocarbons in ambient air [2005] OJ L23/3.

5. p5 Title subsection 2.3.3 Change Levoglucosna by Levoglucosan

Response: The type will be corrected to "2.3.3 Levoglucosan, mannosan and galactosan".